# Inactivation of *E. coli*, *S. aureus,* and Bacteriophages in Biofilms by Humidified Air Plasma

**DOI:** 10.3390/ijms23094856

**Published:** 2022-04-27

**Authors:** Xinni Liu, Zhishang Wang, Jiaxin Li, Yiming Wang, Yuan Sun, Di Dou, Xinlei Liang, Jiang Wu, Lili Wang, Yongping Xu, Dongping Liu

**Affiliations:** 1School of Bioengineering, Dalian University of Technology, Dalian 116024, China; xinni@mail.dlut.edu.cn (X.L.); wanglili@dlut.edu.cn (L.W.); xyping@dlut.edu.cn (Y.X.); 2School of Electrical Engineering, Dalian University of Technology, Dalian 116024, China; wangzhishang@mail.dlut.edu.cn (Z.W.); Dec18_Lee@mail.dlut.edu.cn (J.L.); wymyiming666@mail.dlut.edu.cn (Y.W.); SY9901@mail.dlut.edu.cn (Y.S.); Doudi0412@163.com (D.D.); lxl595595202204@163.com (X.L.); a846903248@163.com (J.W.)

**Keywords:** biofilms, disinfection, bacteriophage, plasma inactivation, *E. coli*, proteins, dielectric barrier discharge, humidified air plasma

## Abstract

In this study, humidified air dielectric barrier discharge (DBD) plasma was used to inactivate *Escherichia coli* (*E. coli*), *Staphylococcus aureus* (*S. aureus*), and bacteriophages in biofilms containing DNA, NaCl, carbohydrates, and proteins. The humidified DBD plasma was very effective in the inactivation of microbes in the (≤1.0 μm) biofilms. The number of surviving *E. coli*, *S. aureus*, and bacteriophages in the biofilms was strongly dependent on the constituent and thickness of the biofilms and was greatly reduced when the plasma treatment time increased from 5 s to 150 s. Our analysis shows that the UV irradiation was not responsible for the inactivation of microbes in biofilms. The short-lived RONS generated in the humidified air DBD plasma were not directly involved in the inactivation process; however, they recombined or reacted with other species to generate the long-lived RONS. Long-lived RONS diffused into the biofilms to generate very active species, such as ONOOH and OH. This study indicates that the geminated NO_2_ and OH pair formed due to the homolysis of ONOOH can cause the synergistic oxidation of various organic molecules in the aqueous solution. Proteins in the biofilm were highly resistant to the inactivation of microbes in biofilms, which is presumably due to the existence of the unstable functional groups in the proteins. The unsaturated fatty acids, cysteine-rich proteins, and sulfur–methyl thioether groups in the proteins were easily oxidized by the geminated NO_2_ and OH pair.

## 1. Introduction

Very recently, the humidified air plasma has been found to generate a strong inactivation effect, with the microbe inactivation directly correlating with the humidity content [1,2,3]. Dielectric barrier discharge (DBD) plasma with a circulation system may disinfect the whole space with its size of 50 × 50 × 70 cm^3^, indicating that humidity accelerates the disinfecting power by generating reactive species from water molecules as well as delivering them through water vapor [1]. The addition of water vapor into the air DBD plasma strongly increases the inactivation efficiency of *Bacillus subtilis* (*B. subtilis*) spores from less than one order of magnitude in dry air up to approximately two orders of magnitude at 80% relative humidity [2]. The air/vapor DBD plasma demonstrates significant antibacterial properties, leading to a decrease in the activation of foodborne pathogens by a factor of 35.5 for *Listeria monocytogenes*, 166 for *Salmonella Typhimurium*, and 266 for *Escherichia coli* within 15 s [3]. Both the air DBD in direct contact with the substrate and remote plasma treatment sources are used for a comparison on the effectiveness of surface decontamination against feline calicivirus and *Salmonella spp.*, and the decontamination efficacy is enhanced by the presence of humidity on the sample surface [4]. The reactive oxygen and nitrogen-containing species (RONS) from the air DBD plasma dissolve in water and react with each other, leading to a significant decrease in *Aspergillus flavus* metabolic activity and spore counts, with the maximal reduction of 2.2 Log units [5].

No positive correlation between UV irradiance or ozone concentration and spore inactivation has been observed for the humidified DBD plasma inactivation, and other reactive species formed from water molecules, such as OH, H_2_O_2_, and nitrogen-containing species can play an important role in the rapid spore inactivation [2]. At high humidity (70–90%), the concentration of OH radicals and H_2_O_2_ is greatly increased while the quantity of generated zone is reduced [1]. The co-existence of O_3_ and NO_2_ in the remote air DBDs enhances the inactivation, and a chemical kinetics model indicates a correlation between the gas-phase N_2_O_5_ concentration and the inactivation of viruses [4]. N_2_O_5_ can either react with the gas-phase H_2_O due to the evaporation of the water droplets or dissolve in the liquid phase to generate peroxynitrous acid (ONOOH) [4]. Chemical and optical measurements indicate that various RONS, such as H_2_O_2_, NO_2_^−^, NO_3_^−^, OH, ONOOH, and ^1^O_2_, are formed in water due to the interaction between the air plasma and water, leading to the bactericidal effect [6]. ONOOH produced via the reaction of HNO_2_ with H_2_O_2_ in the aqueous solution was shown to significantly participate in the antibacterial properties, and it could react with phenol to generate nitrated and nitrosylated products of phenol and hydroxylated products [7]. The decomposition efficiency of gaseous naphthalene in O_2_/H_2_O DBD plasma at a humidity level ~10% was lower than that obtained in air/H_2_O (~28%) DBD plasma [8]. The OH radicals produced from the decomposition of water molecules could play a crucial role in the degradation of naphthalene and inactivation of microbes [8,9]. However, the inactivation mechanism of microbes in the humidified air plasma remains unclear so far.

Airborne microbes are usually encased within self-produced extracellular polymeric substances (EPS), which include a range of biopolymers, such as protein, polysaccharides, DNA, RNA, lipids, surfactants, and water [10,11]. The EPS material restricts the mobility of microbes, keeping them in close proximity, which results in cell–cell communication, horizontal gene transfer, and the formation of synergistic micro consortia. The biofilms including microbes and EPS are attached to a living or non-living surface, and they are typically several microns thick [11]. Biofilms are often highly resistant to the inactivation of microbes. The evaporation and dispersion of sputum droplets from human coughs or sneezes can also contaminate the surface of objects; 10 micron sputum droplets will evaporate to become a droplet nucleus (3.5 microns) in 0.55 s at a humidity level of 80% and in 0.25 s at a humidity level of 20% [12]. Constituents of dried droplet nuclei containing microbes, NaCl, proteins, carbohydrates, and lipids become biofilms on the contaminated surface. The formation of dense biofilms can contribute to tolerance to antimicrobials. The main motivation for this work comes from great concern about airborne transmission of severe acute respiratory syndrome coronavirus 2 (SARS-CoV-2) to the health care community and the effect of biofilms on the plasma inactivation of microbes. Bio-aerosols such as sputum droplets have been referred to as an important mode of transmission of coronavirus disease (COVID-19) [13]. The main objectives of this work are to analyze the effect of biofilms on the plasma inactivation efficiency and to evaluate the inactivation of microbes in biofilms by humidified air plasma with a special emphasis on the humified air plasma inactivation mechanism.

## 2. Results

The steel discs with dry biofilms containing microbes, proteins, carbohydrates, DNA, and salt were placed into the discharge chamber with the inoculated side upwards (Figure 1a). The thickness of the biofilms on the steel discs varied from 0.05 μm to ~2.0 μm. The steel disc with the dry biofilm had its back to the surface DBD plasma to avoid UV irradiation. At the peak-to-peak voltage (V_PP_) of 8.25 kV, the humidified air DBD plasma was generated over the ceramic plate with a surface area of 155 mm × 155 mm (Figure 1b). The discharge power almost linearly increased from 25 W to 710 W when V_PP_ increased from 5.7 kV to 9.2 kV (Figure 1c). In this study, the plasma inactivation of microbes in biofilms was performed at V_PP_ = 8.25 kV, resulting in a discharge power of 460 W.

Figure 2 shows the number of surviving *Escherichia coli* (*E. coli*), *Staphylococcus aureus* (*S. aureus*), and bacteriophages in protein biofilms after exposure to plasma as a function of treatment time. The plasma treatment of 10 s resulted in the complete inactivation of *E. coli* not covered with biofilm. This confirms that humified air DBD plasmas are very effective in the inactivation of *E. coli* [1,3]. However, the inactivation efficiency of *E. coli* in protein biofilms was greatly reduced with an increasing thickness of the protein film (Figure 2a). After the plasma treatment of 20 s, the number of surviving *E. coli* in protein biofilms with a thickness of 0.05 μm, 0.25 μm, 0.5 μm, and 1.0 μm decreased by 2.2 Log, 1.0 Log, 0.8 Log, and 0.75 Log, respectively (Figure 2a). At a given thickness of protein biofilms, increasing the treatment time from 0 to 150 s led to a significant decrease in the number of surviving *E. coli*, indicating that the humified air DBD plasma was very effective in the inactivation of *E. coli* in protein biofilms.

Similar to the inactivation of *E. coli* not covered with proteins, the number of surviving *S. aureus* decreased by >4 Log within 10 s (Figure 2b). The humidified air DBD plasma was very effective in the inactivation of Gram positive and negative bacteria. After the plasma treatment of 20 s, the number of surviving *S. aureus* in protein biofilms with a thickness of 0.05 μm, 0.25 μm, 0.5 μm, and 1.0 μm decreased by 2.3 Log, 0.7 Log, 0.6 Log, and 0.5 Log, respectively. This is consistent with the inactivation efficiency of *E. coli* in protein biofilms. At a given thickness of protein biofilms, the number of surviving *S. aureus* was greatly reduced with increasing treatment time. The bacteriophages not covered with protein biofilms were completely inactivated within 20 s, as shown in Figure 2c. After the plasma treatment of 40 s, the number of surviving bacteriophage in protein biofilms with a thickness of 0.05 μm, 0.25 μm, 0.5 μm, and 1.0 μm decreased by 2.2 Log, 1.4 Log, 1.0 Log, and 0.8 Log, respectively (Figure 2c). At a fixed thickness of the protein biofilms, the inactivation efficiency of bacteriophage significantly decreased when the treatment time increased from 5 s to 150 s. Although the inactivation efficiency of microbes in the protein biofilms was mainly affected by the thickness of the protein biofilms, the humified air plasma was very effective in the inactivation of *E. coli*, *S. aureus*, and bacteriophages in the (≤1.0 μm) protein biofilms.

The effects of the main constituents of biofilms, such as proteins, carbohydrates, DNA, and salt on the plasma inactivation of *E. coli* and bacteriophages were compared, as shown in Figure 3 and Figure 4. The number of surviving *E. coli* in 0.04 μm DNA, 0.3 μm NaCl, and 0.68 μm carbohydrate decreases by ~3 Log within 40 s, and they showed similar tolerance to the plasma inactivation (Figure 3). Figure 4 shows that the plasma treatments for 40 s led to the complete inactivation of bacteriophage in 0.04 μm DNA, 0.3 μm NaCl, or 0.68 μm carbohydrate. Constituents of biofilms, such as DNA, NaCl, and carbohydrate are not highly resistant to the inactivation of *E. coli* and bacteriophages in biofilms. After the plasma treatment time of 40 s, the number of surviving *E. coli* and bacteriophages in the 1.15 μm protein biofilms decreased by 0.9 Log and 0.3 Log, respectively. The inactivation efficiency of *E. coli* and bacteriophages in the 1.15 μm protein biofilm slowly decreased when the treatment time increased from 5 s to 150 s. The 1.15 μm protein biofilm was highly resistant to the plasma inactivation of *E. coli* and bacteriophages compared to 0.04 μm DNA, 0.3 μm NaCl, and 0.68 μm carbohydrate. After plasma treatment for 40 s, the number of surviving *E. coli* and bacteriophages in the 1.87 μm mixed biofilms containing 2% DNA, 19% NaCl, 29% carbohydrates, and 50% proteins decreased by 0.7 Log and 0.3 Log, respectively, indicating that the high resistance to the plasma inactivation of *E. coli* and bacteriophages was mainly affected by the 50% proteins in the biofilms. Our measurements show that the inactivation efficiency of *S. aureus* as a function of time is consistent with that of *E. coli* and bacteriophages, which strongly depended on the thickness of the protein biofilms. However, it was not significantly affected by 0.04 μm DNA, 0.3 μm NaCl, or 0.68 μm carbohydrate.

Three mL of terephthalic acid (TA) or 2,7-dichlorodihydrofluorescein diacetate (H2DCFDA) solutions as chemical probes was spread onto the petri dish to capture the RONS from the gas phase (Figure 1a). Both TA and H2DCFDA tended to react with RONS in solution, and their oxidation products can be identified by fluorescent measurements [14,15]. However, their molecular structures were completely different. A comparison between their oxidation pathways will be crucial for evaluating the chemical reactivity of RONS, such as ONOOH. Terephthalic acid (TA) can be oxidized into 2-hydroxyterephthalic acid (HTA) by RONS in aqueous solution, which emit light at λ = 425 nm (Figure 5a) [14]. The fluorescence intensity of HTA in the aqueous solution significantly increased with an increase in the plasma treatment time. The surface density of TA oxidized into HTA was calculated as a function of the treatment time (Figure 5b). When the treatment time varied from 5 s to 60 s, the surface density of HTA significantly increased, almost linearly, from 5.3 nM/cm^2^ to 59 nM/cm^2^, leading to the oxidation rate of 1.0 nM/cm^2^·s.

After the deacetylation of H2DCFDA in the NaOH solution, the non-fluorescent form (2,7-dichlorodihydrofluorescein, DCFH) can be oxidized into the highly fluorescent form 2,7-dichlorofluorescein (DCF) by the RONS from the gas phase [15]. The fluorescence spectra of DCF at 521 nm are shown in Figure 6. The fluorescence intensity of DCF in the aqueous solution significantly increased with an increase in the treatment time, confirming that DCFH was oxidized into DCF by the RONS from the gas phase.

## 3. Discussion

It was previously reported that *E. coli* and spore inactivation was closely related to the humidity content in the gas phase, indicating that RONS formed from water molecules played a crucial role in the plasma inactivation process [2,3]. Our study also showed that the humidified air DBD plasma was very effective in the inactivation of Gram positive and negative bacteria and bacteriophages in biofilms. Ozone (O_3_) was produced in the humified air DBD plasma, and it can dissolve into water and react with the proteins and lipids of microbes, which contain unstable functional groups [16]. Mass spectrometry analysis indicated that the amino acid residues located on all three chains and the main structural parts of the protein were involved in the ozone-induced oxidation [17]. The O_3_ attack of proteins was directed toward the monomeric units containing tryptophan, methionine, cysteine, and tyrosine [18,19]. The general amino acid reactivity towards ozone was tryptophan > methionine > cysteine > tyrosine > phenylalanine [19]. However, the concentration of O_3_ in the humidified air plasma was decreased by 40% compared to that at a relatively low humidity level, indicating that the very effective inactivation of microbes was not closely related to the O_3_ oxidation [1]. This can be due to the low O_3_ solubility, the slow rate of O_3_ transfer from the gas phase to the liquid, and high O_3_ stability [20]. It has been widely accepted that UV irradiation by the atmospheric-pressure air plasma does not significantly change the bactericidal activity of the plasma [1,2], presumably due to the low power density of UV light from the plasma. In this study, the UV irradiation was not responsible for the effective inactivation of microbes in biofilms since the steel disc with the biofilm had its back to the surface DBD plasma.

The direct air plasma treatment of wet Chinese kale seeds efficiently inactivated *Alternaria brassicicola*, and measurements indicated that both long-lived RONS species and short-lived oxygen-containing species such as OH, O, and HO_2_ could be crucial for the inactivation of microbes [21]. The humidified air DBD plasma accelerates the generation of OH, HO_2_, H_2_O_2_, and HNO_2_ in the gas phase [3]. OH radicals can be produced via the collisions of H_2_O molecules with electrons [3,22]:H_2_O_(g)_ + e → OH_(g)_ + H^+^_(g)_ + 2e(1)
H_2_O_(g)_ + e → OH_(g)_ + H^−^_(g)_(2)

OH radicals can be also formed via the reaction of O(^1^D) with water molecules:O_2(g)_ + e → O(^1^D) _(g)_ + O_(g)_ + e(3)
O(^1^D) _(g)_ + H_2_O_(g)_ → 2OH_(g)_(4)

OH radicals can react with O_3_ to generate HO_2_ radicals:OH_(g)_ + O_3(g)_ → HO_2(g)_ + O_2(g)_(5)

However, the lifetimes of OH and HO_2_ radicals are so short that they do not diffuse into the biofilms. They are mainly formed in the discharge regions, but they can contribute to the formation of long-lived RONS, such as H_2_O_2_ and HNO_2_, as shown below
OH_(g)_ + OH_(g)_ →H_2_O_2(g)_(6)
NO_(g)_ + OH_(g)_ → HNO_2(g)_(7)

Various water clusters Y(H_2_O)_m_ (m < 20–30), where Y is HNO_2_, H_2_O_2_, HNO_3_, OH, HO_2_, ONOOH, NO_x_, O_2_^−^ or others could be formed during the humidified air discharge [23]. The water clusters are relatively stable, and they can be very helpful for transferring RONS onto the biofilms. Water clusters containing ONOOH can be formed in humidified air plasma via the reactions described below [24,25].
OH_(aq)_ + NO_2(aq)_ → ONOOH_(aq)_(8)
HO_2(aq)_ + NO_(aq)_ → ONOOH_(aq)_(9)
O_2_^−^_(aq)_ + NO_(aq)_ → ONOO^−^_(aq)_(10)
ONOO^−^_(aq)_ + H^+^_(aq)_ → ONOOH_(aq)_(11)

However, ONOOH can be quickly transferred into HNO_3_, and the lifetime of ONOOH molecules in the aqueous solution can be only several ms [26]. The peroxidic O-O bond is much weak in ONOOH and thus is very active due to O-O bond homolysis. ONOOH can decompose into NO_2_ and OH with 30% radical yield of homolysis [26]. The homolysis of ONOOH into NO_2_ and OH radicals does not occur in one step, and the initial product is a geminated OH and NO_2_ pair in [ONO⋅⋅⋅OH]_cage_ [26]. The geminate pair can diffuse out of the solvent cage into the solution to become radical or it can collapse in the cage to form HNO_3_ or other end products.
ONOOH ↔ [ONO⋅⋅⋅OH]_cage_↔NO_2_ + OH(12)

NO_2_ and OH radicals can quickly recombine or react with other species to become more stable in aqueous solution. Therefore, NO_2_ and OH radicals usually exist in the solvent cage or in the form of a geminate pair during the homolysis of ONOOH. ONOOH can be generated via the reactions occurring between the long-lived RONS, as shown below [24,25].
N_2_O_5(aq)_ + H_2_O_(aq)_ → 2ONOOH_(aq)_(13)
NO_(aq)_ + NO_2(aq)_ + H_2_O_(aq)_ → 2HNO_2(aq)_(14)
H_2_O_2(aq)_ + HNO_2(aq)_→ONOOH_(aq)_ + H_2_O_(aq)_(15)

Our measurements by chemical probes showed that in the aqueous solution, TA and DCFH can be oxidized into TAH and DCF, respectively, as shown in Figure 7 and Figure 8. During the TA oxidation, one H atom in TA was replaced by one OH and NO_2_ group in the solvent cage to generate 2-hydroxyterephthalic acid (HTA) and 2-nitroterephthalic acid, respectively (Figure 7). The oxidation of each TA by ONOOH leads to the formation of a HNO_2_ or H_2_O molecule in the solution. After being excited at λ = 310 nm, HTA emits at λ = 425 nm. During the DCFH oxidation, two H atoms were removed from each DCFH, leading to the formation of a larger delocalized π bond (Figure 8). The oxidation of DCFH leads to the formation of a H_2_O and HNO_2_ molecule in the solution. This indicates that RONS, such as HNO_2_, H_2_O, and ONOOH, can dissolve and penetrate into water to react with TA and DCFH, which is involved in long timescale chemistry. The short-lived RONS, such as OH, O, HO_2_, O_2_^−^, O_3_^−^, e, generated in the humidified air plasma will not be directly involved in the inactivation of microbes in biofilms. However, their subsequent reactions in the gas phase and water clusters may lead to the formation of long-lived RONS, such as O_3_, NO_x_, HNO_2_, HNO_3_, and H_2_O_2_ [22,24,25]. NO_x_, HNO_2_, HNO_3_, and H_2_O_2_ can diffuse into the biofilms and react with each other to generate very active species, such as ONOOH and OH (Figure 9).

The reaction of HNO_2_ with H_2_O_2_, leading to the formation of ONOOH, is affected by the pH value of aqueous solutions [7]. At pH 3.3, the rate of ONOOH formation was in the range of 10^−8^–10^−9^ M s^−1^. The ONOOH was shown to oxidize phenol into its nitrated and nitrosylated products and significantly participates in the antibacterial properties of PAW. One geminated NO_2_ and OH pair in the [ONO⋅⋅⋅OH]_cage_ could simultaneously attack one TA or DCFH in the aqueous solution, leading to the synergistic oxidation reaction process (Figure 7 and Figure 8). Due to the difference in their chemical stability, the rates of TA and DCFH oxidized by ONOOH were different in the aqueous solution.

Our study shows that the proteins are directly involved in the protection against the inactivation of microbes in biofilms due to the humidified air DBD plasma. Calculation of the highest occupied molecular orbital (HOMO) energy for amino acids showed that unsaturated fatty acids, cysteine-rich proteins, and sulfur–methyl thioether groups in the proteins were very reactive, and they are easily oxidized by oxidants [27]. They can quickly react with the geminated NO_2_ and OH pair in the [ONO⋅⋅⋅OH]_cage_, leading to a decrease in the density of ONOOH in the biofilms (Figure 9). Thus, the protein biofilm is effective in protecting microbes against the inactivation process. The carbohydrate contains no unsaturated functional groups, which are relatively stable during the oxidation. RNA biomacromolecules containing the unsaturated functional groups could be easily oxidized by the geminated NO_2_ and OH pair; however, their concentration in the biofilms was relatively low.

## 4. Materials and Methods

### 4.1. Plasma Inactivation System

The plasma inactivation system consists of a tank filled with distilled water, a discharge chamber, surface DBD device, and a power supply, as shown in Figure 1a. The discharge chamber is made of stainless steel, and it measures 40 cm × 30 cm × 20 cm. The surface DBD device consists of a layer of aluminum tape (0.13 mm thick) serving as an H.V. electrode, an insulating alumina plate (1.0 mm × 175 mm × 175 mm), and a patterned ground electrode (30 μm thick) covered by a layer of Kapton polyimide tape. An AC power source was capable of generating the 1.0–10.0 kV peak-to-peak voltage (V_PP_) with a frequency of 10 kHz. The surface DBD plasma was generated over the alumina plate with a surface area of 155 mm × 155 mm. One capacitor of 2.2 μF in series to ground was used to obtain a Lissajous figure of this charge. Measurements for the charges across this capacitor and the applied voltage across the DBD device resulted in the Lissajous figure, which was used to calculate the discharge power. There was an ultrasonic mist maker in the tank filled with distilled water. The fan supplied a small amount of air to the tank (~1.5 L per minute). The air was humidified when passing over the water surface, and the humidity level was 60–70% at a room temperature of 20 °C. The humified air entered the discharge chamber for the inactivation of microbes in biofilms or the measurements of RONS during the DBD operation. The voltage, current, and charge measurements were conducted during the sterilization experiments. The inactivation of microbes in biofilms was performed at V_PP_ = 8.25 kV, resulting in the discharge power of 460 W. The concentration of ozone in the discharge chamber was monitored during the DBD operation, and it was ~250 ppm.

### 4.2. Preparation of Dry Biofilms on Steel Discs

We prepared the aqueous solutions containing the main constituents of biofilms (i.e., water, NaCl, carbohydrates, protein, and DNA). Lipids were not added to the solutions since they are insoluble in water. Table 1 shows the composition and volume fraction of aqueous solutions and molecular weights and densities of each constituent. The concentration of proteins in aqueous solutions varied from 0 to 23.25 mg/mL. The maximum concentrations of carbohydrates, DNA, and salt were 13.5 mg/mL, 0.834 mg/mL, and 9.0 mg/mL, respectively; 100 mL of aqueous solutions were added to 100 mL of fresh *E. coli* (~10^8^ CFU/mL), *S. aureus* (~10^8^ CFU/mL), and bacteriophage (~10^8^ CFU/mL) cultures to obtain the aqueous solutions containing microbes and the main constituents of biofilms. Ten μL of the aqueous solutions was spread onto each steel disc (1.0 cm^2^) and placed for 10 min under air flow in a biosafety cabinet to allow drying. The steel discs with dry biofilms were placed into the humidified air DBD chamber for plasma inactivation. The thickness of the dry biofilms containing proteins, carbohydrates, DNA, and salt was calculated according to the concentrations and density of each constituent. The steel disc with dry biofilms was placed above the surface DBD plasma (3.0 cm) in the vicinity of the gas outlet. The steel disc with the dry biofilm had its back to the surface DBD plasma to avoid UV irradiation. Plasma inactivation of microbes in the biofilms was performed by changing the treatment time.

### 4.3. Culture Method

After plasma treatment, each steel disc was put into the centrifuge tube containing 2 mL deionized water for 5 min ultrasonication. Then, 300 μL of bacterial suspensions was spread onto the LB agar medium. After overnight culture at 37 °C, colony counting was performed to determine the colony units per cm^2^. Then, 200 μL of bacteriophage suspensions was mixed with 300 μL host bacteria and 5 mL semi-solid agar before being placed on the LB agar medium. Circular, clear, and transparent plaques of the phage appeared after a 4 h culture at 37 °C. Data represent the mean and standard deviation of three independent biological replicates. The statistical significance of differences between all replicate samples was determined by using STDEVP analysis in Excel, and the figures were drawn by Origin software. The significant level of differences was set as *p* < 0.05.

### 4.4. RONS Measurement by the Chemical Probe of Terephthalic Acid (TA)

Terephthalic acid (TA) can be oxidized into 2-hydroxyterephthalic acid (HTA) by RONS in aqueous solution, which can be identified by fluorescence measurement [14]. When the TA/HTA solution is irradiated by UV light (λ = 310 nm), HTA molecules emit light at λ = 425 nm; however, TA molecules do not. TA (MΛCKLIN) does not dissolve in acidic/neutral liquid and thus an aqueous solution of TA was prepared by dissolving TA in NaOH solution. The initial concentrations of TA and NaOH were 4 nM and 10 nM, respectively. Three mL of TA solution was spread onto the petri dish (64 cm^2^) to capture the RONS from the gas phase. The fluorescence measurements were performed by using a fluorescence spectrometer (Cary Eclipse 2018A43C). To quantify the concentration of TA molecules oxidized into HTA, a calibration curve was obtained by using the standard HTA solution (MΛCKLIN).

### 4.5. RONS Measurement by the Chemical Probe of 2,7-Dichlorodihydrofluorescein Diacetate (H_2_DCFDA)

The 2,7-dichlorodihydrofluorescein diacetate (H2DCFDA) represents a non-fluorescent form of the dye. After deacetylation in the NaOH solution, the non-fluorescent form (2,7-dichlorodihydrofluorescein, DCFH) can be oxidized into the highly fluorescent form 2,7-dichlorofluorescein (DCF) by the RONS in the aqueous solution [15]. The process of deacetylation was provided by hydrolysis with NaOH by mixing 0.5 mL of stock H2DCFDA solution (1 mM prepared in ethanol, Shyuanye company, Shanghai, China) with 2 mL of 0.1 M NaOH solution. The reaction was stopped after 30 min incubation at room temperature by adding 7.5 mL of 0.1 M phosphate buffer. By this procedure, 3 mL of 50 μM working solution of DCFH was obtained and spread onto the petri dish (64 cm^2^) to capture the RONS from the gas phase. The fluorescence measurements were performed by using a fluorescence spectrometer (Cary Eclipse 2018A43C) at an excitation wavelength of 495 nm and observing the fluorescence at 521 nm.

## 5. Conclusions

This study shows that the inactivation efficiency of *E. coli*, *S. aureus*, and bacteriophages is strongly dependent on the thickness and constituents of the biofilms. The humidified air DBD plasma can efficiently inactivate microbes in the (≤1.0 μm) biofilms containing DNA, NaCl, carbohydrates, and proteins. Proteins in the biofilm are highly resistant to the inactivation of microbes in biofilms. Our analysis shows that UV irradiation is not responsible for the effective inactivation of microbes, and the short-lived RONS generated in the humidified air plasma will not be directly involved in the inactivation of microbes in biofilms. However, their subsequent reactions in the gas phase and water clusters may lead to the formation of long-lived RONS, which can diffuse into the biofilms to generate short-lived RONS, such as ONOOH and OH. This study also indicates that the geminated NO_2_ and OH pair in [ONO⋅⋅⋅OH]_cage_ can simultaneously attack various unstable organic molecules in the aqueous solution and cause the synergistic oxidation of organic molecules, thus leading to the oxidation of TA and DCFH into TAH and DCF, respectively. The unsaturated fatty acids, cysteine-rich proteins, and sulfur–methyl thioether groups in the proteins can be easily oxidized by the geminated NO_2_ and OH pair in [ONO⋅⋅⋅OH]_cage_, leading to a decrease in the density of ONOOH in the biofilms.

## Figures and Tables

**Figure 1 ijms-23-04856-f001:**
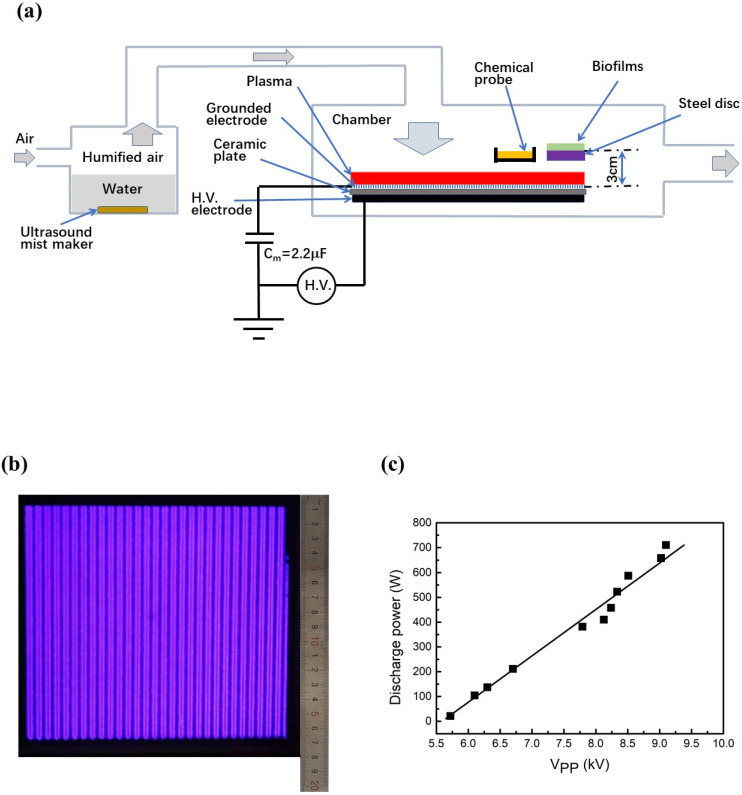
Humidified air surface dielectric barrier discharge (DBD) system for the inactivation of *Escherichia coli* (*E. coli*), *Staphylococcus aureus* (*S. aureus*), and bacteriophages in biofilm. (**a**) Schematic of the humidified air surface DBD plasma system. The plasma inactivation system consists of a tank filled with distilled water, a discharge chamber, surface DBD device, and a power supply. The surface DBD device consists of a layer of aluminum tape (0.13 mm thick) serving as an H.V. electrode, an insulating alumina plate (1.0 mm × 175 mm × 175 mm), and a patterned ground electrode (30 μm thick) covered by a layer of Kapton polyimide tape. An AC power source was capable of generating the 1.0–10.0 kV peak-to-peak voltage (V_PP_) with a frequency of 10 kHz. One capacitor of 2.2 μF in series to ground was used to obtain a Lissajous figure of this charge. The air was humidified when passing over the water surface, and the humidity level was 60–70% at a room temperature of 20 °C. The steel discs with dry biofilms were placed into the humidified air DBD chamber for plasma inactivation. Chemical probes were used to capture the RONS in the gas phase; (**b**) The photo of the surface DBD plasma obtained at V_PP_ = 8.25 kV. The surface DBD plasma was generated over the alumina plate with its surface area of 155 mm × 155 mm; (**c**) The discharge power as a function of V_PP_. Measurements for the charges across this capacitor and the applied voltage across the DBD device resulted in the Lissajous figure, which was used to calculate the discharge power.

**Figure 2 ijms-23-04856-f002:**
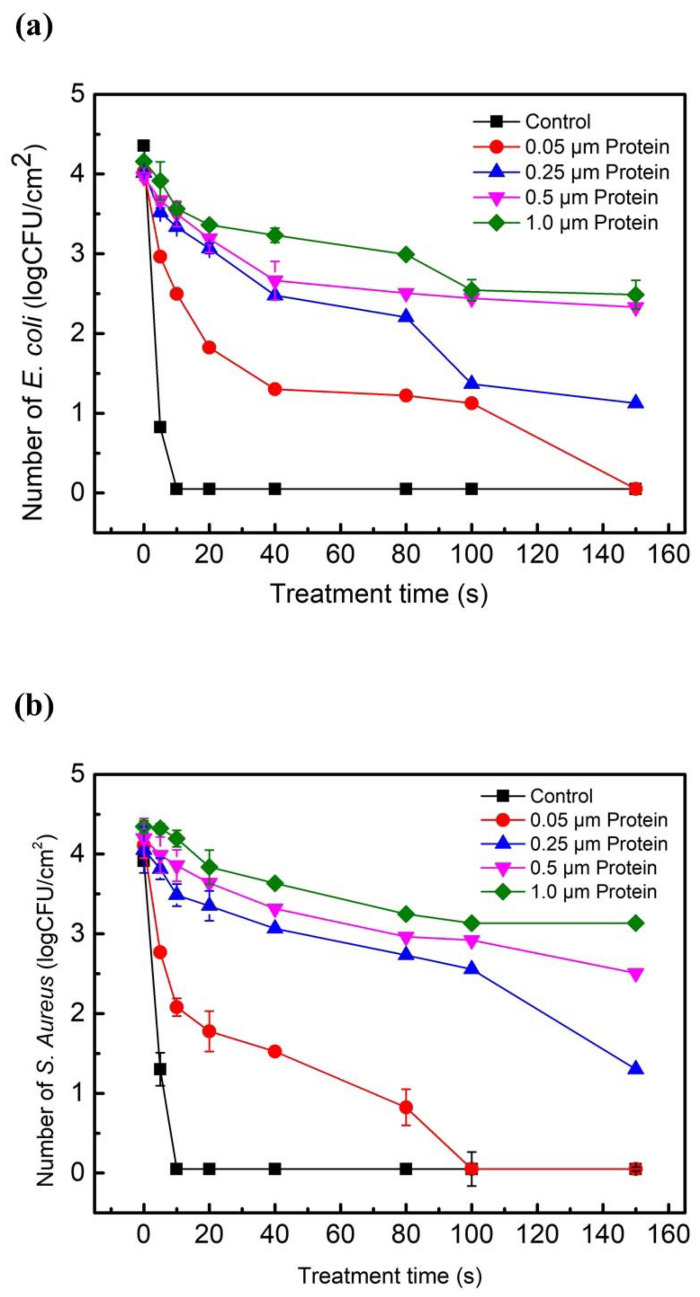
The number of surviving *E. coli* (**a**), *S. aureus* (**b**), and bacteriophages (**c**) in the protein biofilms as a function of the plasma treatment time. The plasma inactivation was performed at V_PP_ = 8.25 kV and a power of 460 W; 10 μL of the aqueous solutions containing microbes and proteins was spread onto each steel disc (1.0 cm^2^) and placed for 10 min under air flow in a biosafety cabinet to allow drying. The thickness of the protein biofilms varied from 0.05 μm to 1.0 μm. The steel discs with dry biofilms were placed into the humidified air DBD chamber for plasma inactivation. The control samples were not covered with protein films. The samples with the treatment time of 0 s were the untreated control. Means ± standard deviations of experiments carried out at least in triplicate are shown.

**Figure 3 ijms-23-04856-f003:**
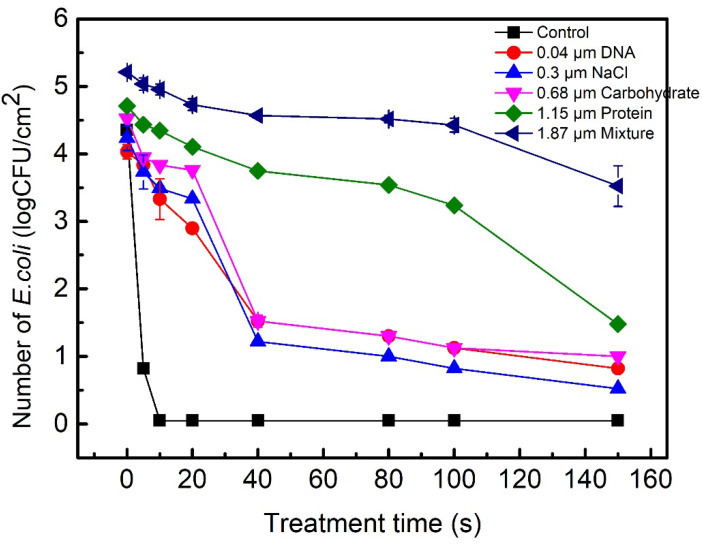
The number of surviving *E. coli* in the biofilms containing proteins, carbohydrates, DNA, salt or their mixture as a function of the plasma treatment time. The plasma inactivation was performed at V_PP_ = 8.25 kV and a power of 460 W; 10 μL of the aqueous solutions containing *E. coli* and constituents of biofilms was spread onto each steel disc (1.0 cm^2^) and placed for 10 min under air flow in a biosafety cabinet to allow drying. After drying, the thicknesses of proteins, carbohydrates, DNA, salt, and their mixed biofilms were 1.15 μm, 0.68 μm, 0.04 μm, 0.3 μm, and 1.87 μm, respectively. The mixed biofilms contained 2% DNA, 19% NaCl, 29% carbohydrates, and 50% proteins. The steel discs with dry biofilms were placed into the humidified air DBD chamber for plasma inactivation. Means ± standard deviations of experiments carried out at least in triplicate are shown.

**Figure 4 ijms-23-04856-f004:**
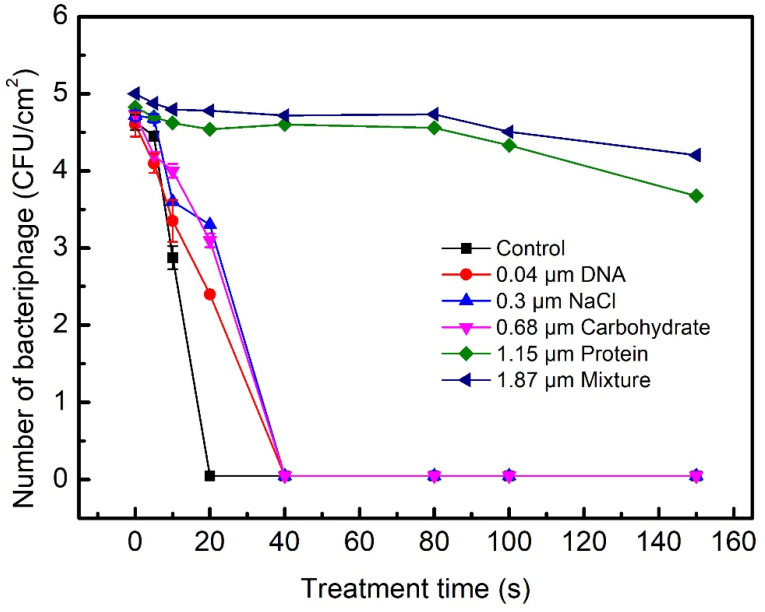
The number of surviving bacteriophages in the biofilms containing proteins, carbohydrates, DNA, salt or their mixture as a function of the plasma treatment time. The plasma inactivation was performed at V_PP_ = 8.25 kV and a power of 460 W; 10 μL of the aqueous solutions containing bacteriophage and constituents of biofilms was spread onto each steel disc (1.0 cm^2^) and placed for 10 min under air flow in a biosafety cabinet to allow drying. After drying, the thicknesses of proteins, carbohydrates, DNA, salt, and their mixed biofilms were 1.15 μm, 0.68 μm, 0.04 μm, 0.3 μm, and 1.87 μm, respectively. The mixed biofilms contained 2% DNA, 19% NaCl, 29% carbohydrates, and 50% proteins. The steel discs with dry biofilms were placed into the humidified air DBD chamber for plasma inactivation. Means ± standard deviations of experiments carried out at least in triplicate are shown.

**Figure 5 ijms-23-04856-f005:**
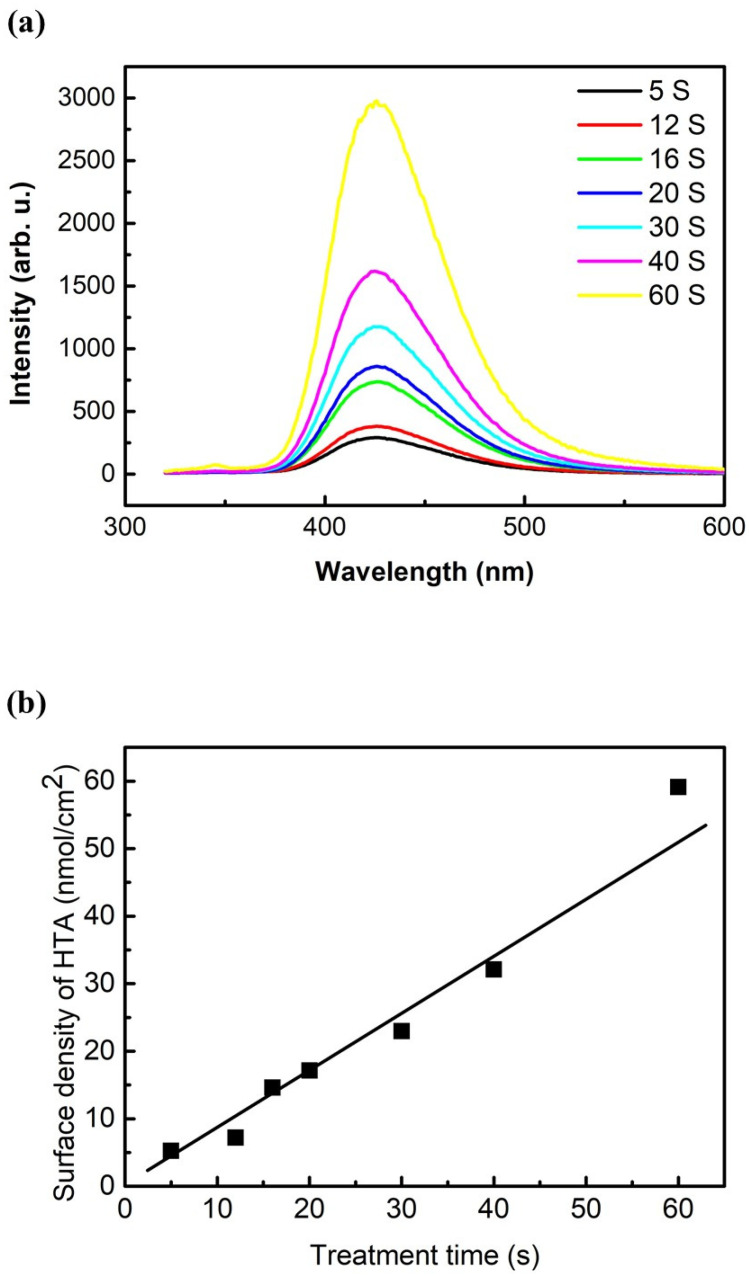
Measurements of RONS in the aqueous solution based on terephthalic acid (TA). Terephthalic acid (TA) was oxidized into 2-hydroxyterephthalic acid (HTA) by RONS in aqueous solution. When the TA/HTA solution was irradiated by UV light (λ = 310 nm), HTA molecules emitted light at λ = 425 nm. (**a**) The fluorescence spectra of the HTA solutions obtained by changing the plasma treatment time. The initial concentrations of TA and NaOH in the aqueous solution were 4 nM and 10 nM, respectively. (**b**) The surface density of HTA obtained in the treated solutions as a function of treatment time. To quantify the concentration of TA molecules oxidized into HTA, a calibration curve was obtained by using the standard HTA solution.

**Figure 6 ijms-23-04856-f006:**
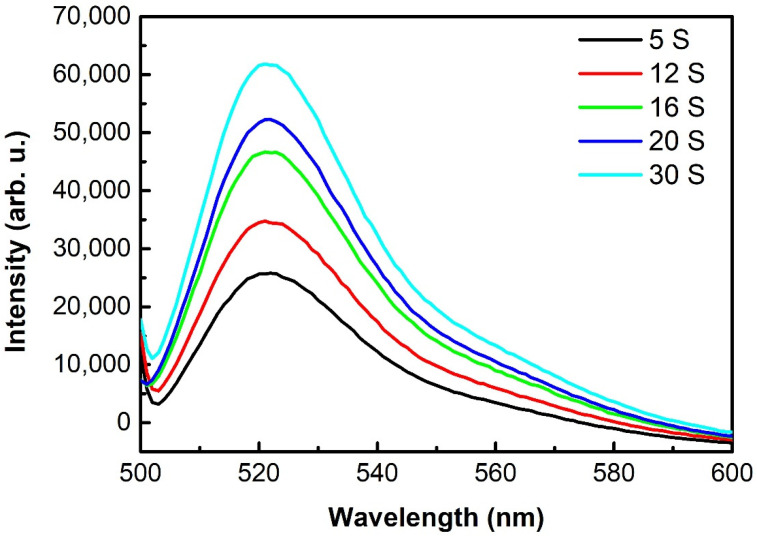
The fluorescence spectra of the 2,7-dichlorofluorescein (DCF) solutions obtained by changing the treatment time; 2,7-dichlorodihydrofluorescein diacetate (H2DCFDA) represents a non-fluorescent form of the dye. After the deacetylation in the NaOH solution, the non-fluorescent form (2,7-dichlorofluorescein, DCFH) was oxidized into the highly fluorescent form 2,7-dichlorofluorescein (DCF) by the RONS in the aqueous solution. The process of deacetylation was shown by hydrolysis with NaOH by mixing 0.5 mL of stock H2DCFDA solution with 2 mL of 0.1 M NaOH solution. The fluorescence was obtained at an excitation wavelength of 495 nm.

**Figure 7 ijms-23-04856-f007:**
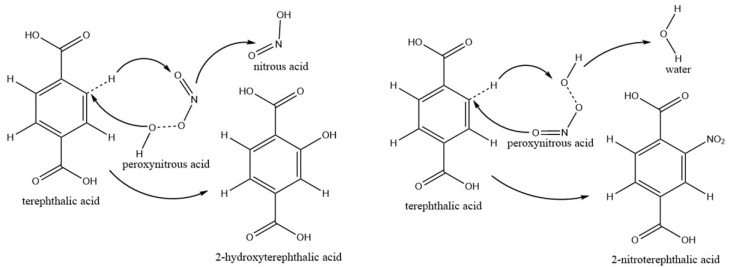
Pathways for the oxidation of terephthalic acid (TA) by ONOOH. The homolysis of ONOOH into OH and NO_2_ radicals leads to the oxidation of TA into 2-hydroxyterephthalic acid or 2-nitrogrephthalic acid.

**Figure 8 ijms-23-04856-f008:**
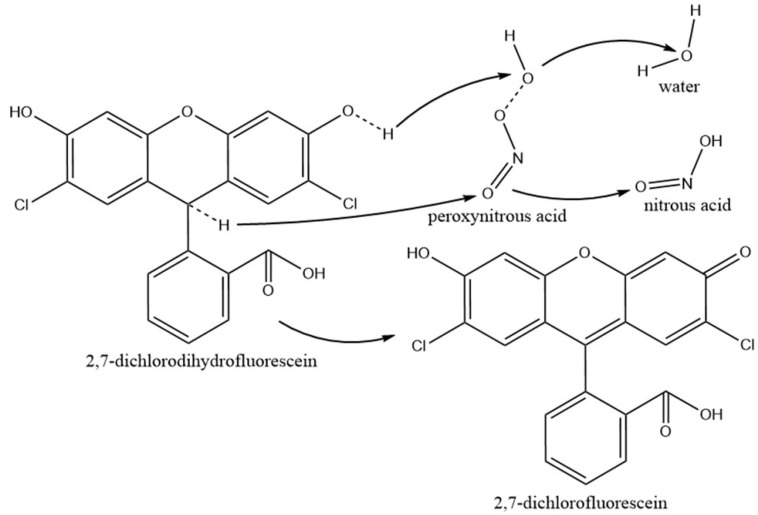
Pathways for the oxidation of 2,7-dichlorodihydrofluorescein (DCFH) by ONOOH. The homolysis of ONOOH into OH and NO_2_ radicals leads to the oxidation of DCFH into 2,7-dichlorofluorescein.

**Figure 9 ijms-23-04856-f009:**
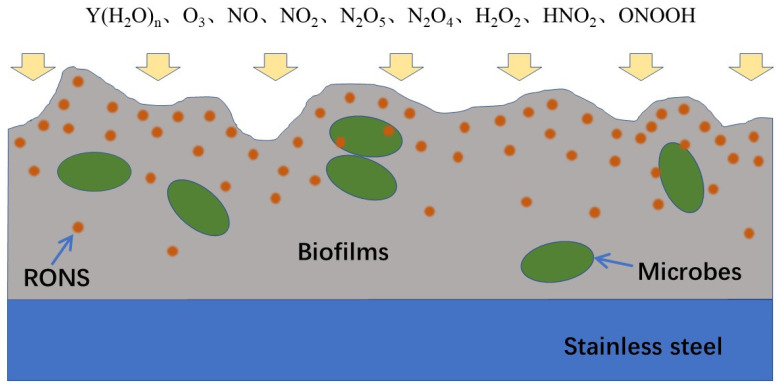
Mechanism for the inactivation of microbes in the biofilms by humidified air plasma. The humidified air plasma generates the long-lived RONS in the gas phase, such as Y(H_2_O)_n_, O_3_, NO, NO_2_, N_2_O_5_, N_2_O_4_, H_2_O_2_, and HNO_2_. These long-lived RONS diffuse into the biofilms and react with each other to generate very active species, such as OH and ONOOH, leading to the oxidation and inactivation of microbes in the biofilms.

**Table 1 ijms-23-04856-t001:** Initial composition and volume fraction of aqueous solutions and molecular weights and densities of each constituent. The concentration of proteins in aqueous solutions varied from 0 to 23.25 mg/mL. The maximum concentrations of carbohydrates, DNA, and salt in aqueous solutions were 13.5 mg/mL, 0.834 mg/mL, and 9.0 mg/mL, respectively. The aqueous solutions were used to prepare the biofilms on steel discs.

Component	Molecular Weight (g/mol)	Density (kg/m^3^)	Concentration (mg/mL)
Water	18	1000	945~1000
Protein	66,500	1300	0, 1.16, 5.8, 11.6, 23.25
Carbohydrate	180.16	1600	0, 13.5
DNA	300	1000	0, 0.834
Salt	58.4	2160	0, 9.0

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
