# Peer review of "Inactivation of *E. coli*, *S. aureus,* and Bacteriophages in Biofilms by Humidified Air Plasma"

_ijms, 2022, doi:10.3390/ijms23094856_

Round 1
Reviewer 1 Report
Review of [ijms-1688691] Manuscript ID: Inactivation of E. coli, S. Aureus and bacteriophage in biofilms by a humidified air plasma
Comments to the Authors:
This article discusses the effects of plasma treatment on E. coli, S. Aureus, and bacteriophage under various types and thicknesses of biofilms. The results of inactivation efficiency are intriguing and will be of great interest to forum readers. After minor revisions, I believe the manuscript is ready for publication in the International Journal of Molecular Sciences.
Although long-lived species dominate the primary interaction between plasma and microorganisms in this work, the plasma component and samples will undoubtedly be involved with both short-lived and long-lived RONS because this work was carried out in highly humidified air and water molecules were involved. The reviewer suggests that the authors refer to the paper (ACS Appl. Mater. Interfaces 2021, 13, 37, 43975–43983 [doi:10.1021/acsami.1c10771]), which discusses the interactions of the plasma discharge on water involving both short-lived and long-lived RONS to deactivate fungus in the literature. It will significantly support both the generation and interaction processes of the short- and long-term RONS generations.
Author Response
请参阅附件。

Reviewer 2 Report
By this paper, authors analyzed the effect of air DBD plasma against E. coli, S. aures and bacteriophage proliferation and survival in a synthetic condition simulating bacterial biofilm.
Different sections of the paper are well described and quite clear, however several typo were detected.
For example:
- Material and methods section refer to table I, however in the table description is written Table 0;
- Material and methods at the line 396 (phosphate buffer and not phosphate butter).
Together with this typo correction, it is strongly suggested to revise the overall paper adding a English spell check to increase fluency of the manuscript.
In the paragraph 4.2 it reported that data representation is by means +/- SE, however it is more correct to indicate standard deviation. Please revise those data including STD.
In the same paragraph, culturing method and conditions are missing, please include them.
FIG.2 a,b and c lack of not treated control. Please add it
FIG.3 Data set for 1.87um mixture is quite far away from the other data. By my opinion there is the need to introduce a parameter which can be used to normalize all the data. Not treated condition can be used for this purpose, for example, in all the experimental conditions (fig2, fig 3, fig 4)
FIG.3/4 Why there is no a figure describing data for S. aureus? There is no mention about missing experiment. Please add it.
It is not very clear why there is the need to use both TA and H2DCFDA instead of only TA excluding H2DCFDA or viceversa. Please, add some sentence to clarify
In my opinion scientific soundness of this paper will be increased by comparing data obtained by DBD plasma with treatment with conventional antibiotics such as gentamicin and vancomycin or genta/vanco synergic activity.
Round 2
Reviewer 2 Report
Paper quality is improved; no other comments from my side.
Thank you